# Limitations of multiexponential $T_1$ mapping of cortical myeloarchitecture

**Jakub Jamárik**[1], **Jiří Vitouš**[2,3], **Radovan Jiřík**[2], **Daniel Schwarz**[4], **Eva Koriťáková**[5]*

**1** Faculty of Medicine, Masaryk University, Brno, Czech Republic, **2** Institute of Scientific Instruments of the CAS, v.v.i., Brno, Czech Republic, **3** Department of Biomedical Engineering, Faculty of Electronics and Communication, Brno University of Technology, Brno, Czech Republic, **4** Department of Simulation Medicine, Faculty of Medicine, Masaryk University, Brno, Czech Republic, **5** Institute of Biostatistics and Analyses, Faculty of Medicine, Masaryk University, Brno, Czech Republic

\* koritakova@iba.muni.cz

## Abstract

Neuropsychiatric malignancies frequently manifest at the level of individual cortical layers. The resolutions currently available for medical magnetic resonance imaging (MRI) prevent the study of these pathologies at clinically available field strengths of 3 T. Previous studies have claimed to have overcome these issues by extensions of quantitative MRI. Following this, the feasibility of multiexponential $T_1$ relaxometry was assessed as a basis for *in vivo* delineation of cortical lamination. Three methods of non-linear least-squares-based multiexponential analysis were examined across key degrees of freedom identified in the literature. The methods employ a wide variety of ways to overcome the common pitfalls of multiexponential analysis, such as regularization, bound constraints, and repeated optimization from multiple starting points. A custom MRI phantom was 3D-printed and filled with various $MnCL_2$ mixtures that represent the spin-lattice relaxation times that commonly occur in neocortical gray and white matter at 3 T. A 96 × 96-voxel image consisting of a single slice was acquired using a FLASH sequence and used to create 10 composite datasets with known distributions of $T_1$ decay constants. The results showed that lowest relative error achieved across multiexponential models was approximately 20%. As achieving even this level of estimation accuracy requires either $T_1$ ratios that rarely occur in the cerebral cortex or knowledge of the number of relaxation components and their expected values to a degree that is seldom feasible, the visualization of cortical layers based on these estimates is unlikely to represent their true distribution. In conclusion, the current methodological approaches do not allow for sufficiently precise estimation of $T_1$ decay constants spanning the range of cortical gray and white matter.

**Data availability statement:** All data are available from Zenodo at https://doi.org/10.5281/zenodo.17456443.

**Funding:** The work on this publication was supported by the Czech Science Foundation (GAČR), URL: https://gacr.cz/en/, project No. GA23-06957S (JJ and DS); by the Masaryk University, URL: https://www.muni.cz/en, project No. MUNI/A/1769/2024 (JJ); by the Czech Science Foundation (GAČR), URL: https://gacr.cz/en/, project No. GA22-10953S (JV and RJ); by the Ministry of Education, Youth and Sports, URL: https://msmt.gov.cz/, project No. LM2023050 (JV and RJ). The funders had no role in study design, data collection and analysis, decision to publish, or preparation of the manuscript.

**Competing interests:** The authors have declared that no competing interests exist.

## Introduction

The human cortex can be parcellated into six layers based on histological studies of its cytoarchitecture [1,2]. Similar laminar composition also manifests when examining the cortical myeloarchitecture [3]. Many neuropsychiatric diseases result from pathological changes at the level of individual cortical layers. Layer-specific pathologies have been observed in schizophrenia [4,5], Alzheimer's disease [6,7], and autism [8]. Unfortunately, these changes cannot be directly observed *in vivo* due to the spatial resolution of current magnetic resonance imaging (MRI) technology.

Conventional 3-T magnetic resonance imaging (MRI) can achieve resolutions of about 0.8–1 mm [9]. The human cortex is on average 2.5 mm thick [10], with up to six cytoarchitectonic layers [1], making the current resolution insufficient. To overcome this limitation, two distinct approaches are currently being explored. The first approach is focused on increasing the spatial resolution, while the second approach involves representing the layers in the spin-lattice relaxation domain.

Increases in spatial resolution are achieved by increasing the MRI field strength from the currently available 1.5 T and 3 T to 7 T and upwards. These field strengths allow for resolutions of 0.7 mm isotropic [11] and in-plane resolution of up to 0.2 mm [9], making cortical layers directly visible. There have been many structural [12] and quantitative applications of cortical imaging at 7 T [13,14].

The high-resolution approach is not without limitations. Relatively low scanner availability prevents effective usage for large-scale population studies. The number of layers that can be delineated at 7 T is not uniform across the brain due to thickness variations of individual layers. The contrast obtained by high-resolution MRI predominantly reflects the myeloarchitecture of the cortical tissue [15], so the observable layers do not directly correspond to the cytoarchitectonic layers known from histology. Additionally, while 7-T scanners generally improve on the partial volume effect (an artifact caused by the averaging of signals emanating from two or more different tissue types in a single voxel), it remains a substantial concern when examining finer details of a given tissue, such as layers of cortical gray matter.

Lifshits et al. proposed an alternative approach for overcoming the insufficient spatial resolution of 3-T MRI, which paradoxically uses low-resolution imaging [16]. The aim of this concept is also to address some of the limitations that cannot be fully overcome even at resolutions of 7 T, such as partial volume artifacts caused by merging multiple cortical layers into a single voxel. Instead of increasing the spatial resolution, the authors attempted to delineate the layers based on their spin-lattice relaxation time estimated at 3 T.

By employing a modification of the conventional $T_1$ relaxometry, multiple $T_1$ times originating from a single voxel could be distinguished. The subsequent classification of these $T_1$ times reflected the structure of the cortical lamina. This idea relies on successful estimation of multiple exponentially decaying components from a single decaying signal, which is known as multiexponential analysis (for further details, see the section Theoretical background of multiexponential analysis). If such an application of multiexponential analysis is to be successful it must respect the fundamental limitations inherent to multiexponential analysis itself.

From a mathematical perspective, multiexponential analysis is an ill-posed problem, particularly when dealing with decay constants of similar values [17]. This makes the solution of the underlying optimization problem, especially in the case of a non-linear multiexponential model, highly sensitive to initial conditions. Various methods have been proposed to address these challenges, and the most common approach involves some form of regularization or optimization constraints [18].

Beyond the theoretical limitations, practical challenges also arise when conducting multiexponential analysis for $T_1$ mapping to delineate cortical layers. Estimating the number of kernel functions $J$ is one such challenge. If the number is not known beforehand, their maximum expected number must at least be specified. A hierarchical fitting strategy is often employed, in which models with an increasing number of kernels are fitted consecutively. The final model is chosen based on a goodness-of-fit measure, such as the sum of squared errors (SSE) [19,20] (This fact is only documented in the actual code provided by the original authors).

Another major difficulty is experimental noise, which exacerbates the already challenging multiexponential analysis. The results of non-linear least-squares-based multiexponential analysis become highly imprecise when the signal-to-noise ratio (SNR) deteriorates to less than 1000:1 (30 dB) [21,22]. This limitation also constrains the number of decay constants that can be accurately estimated, given their ratio. In practice, a minimum ratio of two decay constants of 2.5 is needed, and a ratio of 3.5 is necessary for a confident estimate [21].

We argue that multiexponential $T_1$ relaxometry does not yield sufficient information for the delineation of cortical lamination at 3 T. This is due to its fundamental limitations, which cannot be fully overcome in an MRI setting. We hypothesize that even at the lowest noise levels achievable in practical whole brain MRI settings at 3 T (SNR of 30 dB) [23], the narrow distribution of the $T_1$ relaxation times prevents their sufficiently accurate estimation, thus severely limiting their ability to represent the cortical lamination.

Nevertheless, some works report successful $T_1$ estimation even in the face of these obstacles [16,20]. To test our hypothesis, we analyzed the accuracy of $T_1$ estimation of three methods for multiexponential modeling. We mimicked the distribution of $T_1$ relaxation times in the human cortex by a custom-made MRI phantom with known $T_1$ composition. We also quantified the estimation accuracy across key degrees of freedom that affect the analysis.

## Theoretical background of multiexponential analysis

The term "multiexponential analysis" refers to the estimation of multiple exponentially decaying components from a single decaying signal. It is assumed that a particular signal intensity can be modeled as a sum of exponential functions. The simplest multiexponential model can be postulated as:

$$y(t_i) = \sum_{j=1}^{J} A_j e^{\frac{-t_i}{T_j}},$$

(1)

where $y(t_i)$ is the signal intensity at $N$ discrete times $t_i$, which is dependent on the sum of $J$ components. Each component is an exponentially decaying function. The decay time of each function is denoted by $T_j$, and the amplitude is $A_j$. The exponential term of the function can be considered its kernel. Therefore, multiexponential analysis refers to the estimation of the kernel amplitudes $A_j$ and decay constants $T_j$.

Depending on the studied phenomenon, the kernel can take a variety of different forms. In the case of a signal emanating due to magnetic resonance, the time constant $T$ can correspond to the longitudinal relaxation time $T_1$ or the transverse relaxation time $T_2$. Assuming the phenomenon being studied is entirely dependent on the longitudinal relaxation $T_1$ and assuming an inversion-recovery (IR) acquisition, the model in (1) can take the following form [24]:

$$y(t_i) = \sum_{j=1}^{J} A_j \left(1 - 2e^{\frac{-t_i}{T_{1j}}}\right).$$

(2)

In this model, the term $t_i$ now corresponds to the inversion time $TI_i$, and the decay constant $T_j$ corresponds to the longitudinal relaxation time of the $J$-th component $T_{1j}$. The model in (2) assumes a discrete number $J$ of relaxation components. This assumption could be relaxed, and a continuous distribution of $T_1$ components could be modeled:

$$y(t_i) = \int_a^b A(T_1) \left(1 - 2e^{\frac{-t_i}{T_1}}\right) dT_1,$$

(3)

where $A(T_1)$ is the amplitude spectrum of the $T_1$ distribution. The model in (3) extends the discrete approach by considering a continuous range of relaxation times, thus providing a more comprehensive framework for multiexponential relaxation [25]. It is important to note that this model generalizes to a Fredholm integral equation of the first kind.

There are additional constraints when applying multiexponential analysis to the $T_1$ relaxation signal for MRI data represented as magnitude-only values. While signal intensity modeled using (2) can reach negative values, only positive values are present in data in the case of the magnitude image reconstruction mode. Therefore, either the model must be fitted as an absolute sum of the kernel functions, or the signal must be artificially inverted. This artificial inversion is known as polarity restoration.

## Materials and methods

### Imaging phantom

A physical phantom was constructed by 3D printing a custom-designed model (see supplementary material S1 Fig). The whole phantom consisted of a single solid piece with 19 separate tubular compartments for liquids and six compartments shaped as letters to denote orientation. Seven tubular compartments were filled with different solutions of $MnCl_2$ in saline at concentrations of 4, 2, 1, 1/2, 1/4, 1/8, and 1/16 mM, starting in the middle of the phantom, continuing outwards and clockwise, and filling up the inner ring. The remaining 12 tubular compartments comprising the outer ring were filled with saline.

Each of the solutions exhibited a unique $T_1$ relaxation time ranging from 180 to 2600 ms calculated from measured data (see the section MRI protocol). The number of unique solutions was chosen based on the number of components that could be theoretically encountered in a voxel of the cerebral cortex (the six cortical layers and the cerebrospinal fluid). The relaxation time range was chosen to mimic the values encountered within the human cortex measured by inversion recovery (IR)-based $T_1$ relaxometry [26,27].

### MRI protocol

Images were acquired with a Bruker Biospec 94/30 USR 9.4 T magnet and a BGA12S-HP Gradient insert preclinical MR scanner (Bruker BioSpin GmbH, Ettlingen, Germany). An IR-prepared echoplanar imaging (EPI) sequence was used for $T_1$ quantification of the phantom (a global IR pulse was used based on vendor's FAIR EPI ASL method). The adiabatic inversion pulse was automatically set and verified by the scanner, with the phantom placed in the isocenter to further ensure maximal inversion efficiency. The sequence parameters were the following: TR/TE = 10000/11.02 ms with 50 IR times (one IR time measured per IR pulse) spanning the interval from 100 to 5000 ms spaced equidistantly by 100 ms. A single-slice complex-valued image with dimensions of 96 × 96 voxels was acquired using this imaging protocol. A parametric map of the image was estimated using the built-in functionality in the software Paravision 7 (Bruker BioSpin GmbH, Ettlingen, Germany).

### Ground truth

Several composite datasets were created using the complex single-slice image. A 6 × 6 × 50-voxel window was extracted from each region of interest (ROI), and each ROI represented either one of the seven compartments containing various

MnCl$_2$ solutions or a single compartment with saline. The size of the window was chosen as the size of the largest possible square which could fit inside the circular ROI, excluding the outermost voxels. Three ROIs with insufficient signal quality were excluded from further processing and analysis. The extracted voxels from each ROI had their polarity restored specifically within the third dimension corresponding to the IR times.

The polarity restoration was performed per voxel of extracted ROI as follows. All values preceding the minimum value of the signal were multiplied by −1 (inverted about zero). The minimum value was approximated by the intersection of the real and imaginary parts of the complex signal. The signal was then visually inspected, and if required, individual IR time points were manually corrected.

A series of ROI combinations was defined, based on all combinations possible. All the extracted voxels from the given ROIs were then summed accordingly, yielding 36 multiexponential voxels per combination. Finally, only the magnitude values were retained from the combined data to best simulate the magnitude-only imaging. This process resulted in 10 composite datasets: six datasets consisting of signals from two ROIs, three datasets consisting of signals from three ROIs, and one dataset consisting of signals from four ROIs (see Table 1). The same extraction procedure followed by concatenation instead of summation was applied to the parametric maps to create ground truth for $T_1$ estimation.

## Multiexponential analysis methods

Three multiexponential analysis methods were examined, where each represented one of the two general modeling approaches: linear and nonlinear multiexponential models [18]. The linear modeling approach was represented by the inverse Laplace transform (ILT) method, while the nonlinear approach was represented by bounded optimization with multiple starting points (MUL) and bounded optimization according to Tomer et al. [20] (TOM). All methods have been previously used in the multiexponential analysis of magnetic resonance images either directly to represent cortical lamination [20,22] or in the quantification of multiexponential $T_1$ decay [28]. While the methods differ in several aspects, they all aim to solve the task of multiexponential analysis in the least-squares sense. A breakdown of each method's key features is presented in Table 2.

**Inverse laplace transform.** For the purposes of this work, the inverse Laplace transform was defined as a solution to a special case of the Fredholm integral equation of the first kind in (3) [29]. The multiexponential model of the $T_1$ decay in (3) takes the following form:

$$y\left(TI_i\right) = \sum_j g(T_{1j})\left(1 - 2e^{\frac{-TI_j}{T_{1j}}}\right) + e_i, \tag{4}$$

Table 1. **Overview of the composite datasets. Only four distinct $T_1$ components are listed out of the original seven, due to the poor signal quality of the remaining ROIs.**

| Combination ID | Number of components | First component mean $T_1$ [ms] | Second component mean $T_1$ [ms] | Third component mean $T_1$ [ms] | Fourth component mean $T_1$ [ms] | Mean $T_1$ ratio |
|---|---|---|---|---|---|---|
| Comb-1 | 2 | 641 | 1039 | – | – | 1.62 |
| Comb-2 | 2 | 641 | 1540 | – | – | 2.40 |
| Comb-3 | 2 | 641 | 2733 | – | – | 4.26 |
| Comb-4 | 2 | 1039 | 1540 | – | – | 1.48 |
| Comb-5 | 2 | 1039 | 2733 | – | – | 2.63 |
| Comb-6 | 2 | 1540 | 2733 | – | – | 1.77 |
| Comb-7 | 3 | 641 | 1039 | 1540 | – | 1.55 |
| Comb-8 | 3 | 641 | 1039 | 2733 | – | 2.13 |
| Comb-9 | 3 | 1039 | 1540 | 2733 | – | 1.63 |
| Comb-10 | 4 | 641 | 1039 | 1540 | 2733 | 1.63 |

where $i \in \{1, \ldots, N\}$ is the index of measured inversion times, $j \in \{1, \ldots, J\}$ is the index of possible $T_1$ relaxation times, and $e_i$ is the experimental noise. The spectral function $g(T_{1j})$ represents the weights corresponding to a set of $T_1$ values spanning a predetermined range of candidate $T_1$ values. A set of 100 candidate values spanning the range commonly encountered in $T_1$ relaxometry (0–5000 ms) was used. The selection was based on our previous experimental results. The model weights are obtained by minimizing the following expression:

$$g = \mathrm{arg}min_{g \geq 0} \frac{1}{2} \left| Ag - y \right|^2,$$

(5)

Where $A_{ij} = 1 - 2exp(-TI_i/T_{1j})$ is the transformation kernel, and $y = (y(TI_1), y(TI_2), \ldots y(TI_N))$. The minimization is carried out using the nonnegative least-squares algorithm. The number of estimated $T_1$ components is equivalent to the number of non-zero weighted kernel functions. In practical applications, a higher threshold than zero is used (0.075 in our application, set experimentally). The method was implemented based on a previous study [28] with custom MATLAB code, and the final version is available in a public GitHub repository (see https://github.com/JakubJamarik/multiexponential-t1-mapping).

**Bounded optimization with multiple starting points.** The multiexponential model in (2) can be directly fitted in the least-squares sense with an objective function $F$ taking the following form:

$$F(A_1, A_2, \ldots, A_J, T_{11}, T_{12}, \ldots, T_{1J}) = \sum_{j=1}^{J} \left( y(TI_i) - \left| \sum_{j=1}^{J} A_j \left( 1 - 2e^{\frac{-TI_i}{T_{1j}}} \right) \right| \right)^2.$$

(6)

The model in (6) is fitted as an absolute value, and no polarity restoration is performed. The solution to (6) is obtained by minimization using a suitable solver. The search space is limited by a set of linear bounds.

To prevent the solver from becoming trapped at a local minimum, optimization is initiated from multiple starting points. The final solution is chosen among the multiple outcomes based on a goodness-of-fit measure. The number of fitted kernels $J$ is not directly estimated and must be specified before the model is fitted. The method was implemented using custom MATLAB code by extending the code used in our previous work [22] and is available in a public GitHub repository (see https://github.com/JakubJamarik/multiexponential-t1-mapping).

**TOM-er based bounded optimization.** This approach uses the same objective function (6) and linear bounds as the MUL method but differs in two key aspects: it uses only a single starting point and a deliberately chosen number of components $J$. This choice is based on a sequential fit of a model with $J$ increasing up to a predefined threshold representing the maximum number of components. The best-fitting model is then determined based on SSE.

This method was implemented using original code developed by the authors [20] and is available in a public GitHub repository (see https://github.com/JakubJamarik/multiexponential-t1-mapping). A distinction between the TOM and MUL methods should be noted. While the MUL method selects the best-fitting model to overcome possible convergence to a local optimum, the TOM method chooses the best-fitting model based on the most likely number of components.

**Table 2. Selected key aspects of multiexponential analysis methods.**

| Abbre-viation | What model is the method based on? | Minimum No of components | Maximum No of components | How is No of components chosen? | How is the starting point chosen? | Is polarity restoration required before fitting? |
|---|---|---|---|---|---|---|
| **ILT** | Linear | 0 | N | Thresholding | Fixed | Yes |
| **MUL** | Nonlinear | 1 | N | – | Random | No |
| **TOM** | Nonlinear | 2 | N | SSE | Semi-random | No |

ILT – inverse Laplace transform, MUL – bounded optimization with multiple starting points, TOM – TOMer-based bounded optimization.

## Analysis of $T_1$ estimation

The $T_1$ relaxation times corresponding to the different ROI combinations were estimated using the ILT, MUL, and TOM methods. After fitting the models to each dataset, the relative error for each voxel was computed for each method. Before the actual error rate was computed, the parameter estimates were first aligned to best match the ground-truth values (see supplementary material S2 File). This was a necessary step due to the additive nature of the fitted models. While the coefficients of the models can be estimated, their order does not have to follow the one directly returned by the solver.

Voxel-based relative errors were then used for comparison of the three methods. Their accuracy was assessed across three key degrees of freedom: (i) the ratio of $T_1$ decay constants, (ii) the number of signal components, and (iii) the ability to estimate their count. When examining the ratio of $T_1$ constants, a mean $T_1$ ratio per composite dataset was computed as the mean of $T_1$ ratios for each voxel in the given ROI. A linear regression model was employed to examine the relative error's dependence on the $T_1$ ratio (for the full details see supplementary material S4 File).

To determine how each method handles the identification of the correct number of components, data were treated as if they included an additional component with an $A_0$ value of zero during the estimation of fit accuracy. This allowed for comparison of the given methods on the same composite datasets under two conditions: (i) when the expected number of components is equal to the actual number of components and (ii) when it is larger than the actual number of components.

The second case required addition of an "artificial" ground-truth component with the value of $A_0$ equal to zero so that the estimation accuracy of the "artificial" component could be measured. The MUL method was excluded from the analysis as it currently does not support the estimation of the optimal number of components. The maximum number of the expected components, in case of the TOM method, was adjusted accordingly. In case any method estimated less than the expected number of components the remaining components were treated as having $A_0$ equal to zero. Since the introduction of a ground-truth value equal to zero prevents the computation of relative error, a maximum mean error estimate was modified and computed instead (see supplementary material S3 File).

All data processing steps including the creation of the ground truth and multiexponential parameter estimation were performed using MATLAB (v R2018b). All analyses of $T_1$ estimation were performed using R (v 4.4.3).

## Results

The relative error of $T_1$ estimation varied across modeling methods and numbers of components (see Table 3). The ILT method showed the lowest relative error for the 2-component signal (42.67%) and the highest for the 3-component signal (60.30%), and no consistent trend was observed. The MUL method demonstrated the lowest error for the 2-component signal (22.83%) and the highest error for the 4-component signal (46.81%), and the error rate increased as more components were added. The TOM method exhibited the highest relative error for the 2-component signal (23.53%) and the lowest error for the 4-component signal (20.27%), and the error decreased as more components were added. The relative error of $A_0$ estimation followed similar trends, with an overall lower estimation accuracy. The relative error has increased from 56.43% in case of the 2-component signal to 95.34% in case of the 4-component signal for the ILT method, and from 48.78% to 65.82% for MUL method. The TOM method showed a decrease in relative error from 57.79% in case of the 2-component signal to 29.50% in case of the 4-component signal.

Detailed presentation of absolute values of all parameter estimates across all composite data sets in the form of parametric maps and tables, with their mean values and standard deviations, can be found in supplementary materials S5 and S6 Files respectively.

### Number of components and their ratio

All three methods showed a statistically significant linear dependence of the relative error on the $T_1$ ratio ($p < 0.001$ for ILT, MUL, and TOM). The ILT, MUL, and TOM methods showed decreases in mean relative error by 16.9%, 9.8%, and

**Table 3. Mean relative error of different methods based on the number of components.**

| Parameter type | Number of components | Number of voxels | ILT – relative error [%] | MUL – relative error [%] | TOM – relative error [%] |
|---|---|---|---|---|---|
| $T_1$ | 2 | 216 | 42.67 | 22.83 | 23.53 |
| $T_1$ | 3 | 108 | 60.30 | 43.47 | 22.05 |
| $T_1$ | 4 | 36 | 58.60 | 46.81 | 20.27 |
| $A_0$ | 2 | 216 | 56.43 | 48.78 | 57.79 |
| $A_0$ | 3 | 108 | 85.20 | 62.95 | 48.40 |
| $A_0$ | 4 | 36 | 95.34 | 65.82 | 29.50 |

ILT – inverse Laplace transform, MUL – bounded optimization with multiple starting points, TOM – TOMer-based bounded optimization.

4.2%, respectively, with a unit increase in the $T_1$ ratio (see Fig 1). While a global linear trend was observed, the actual variance remained relatively low with $R$-squared values of 0.194, 0.186, and 0.041 for the ILT, MUL, and TOM methods, respectively.

The local variance in relative error became apparent when comparing the three methods using the $T_1$ ratios at which they exhibited their best performance. While both the ILT and MUL methods achieved their lowest mean relative error at a $T_1$ ratio of 4.3 (14.2% and 8.8%, respectively), this was not the case for the TOM method. This method achieved the lowest mean relative error (9.0%) at a ratio of 1.8. The relative error for the previously best-performing ratio (4.3) was 11.3%.

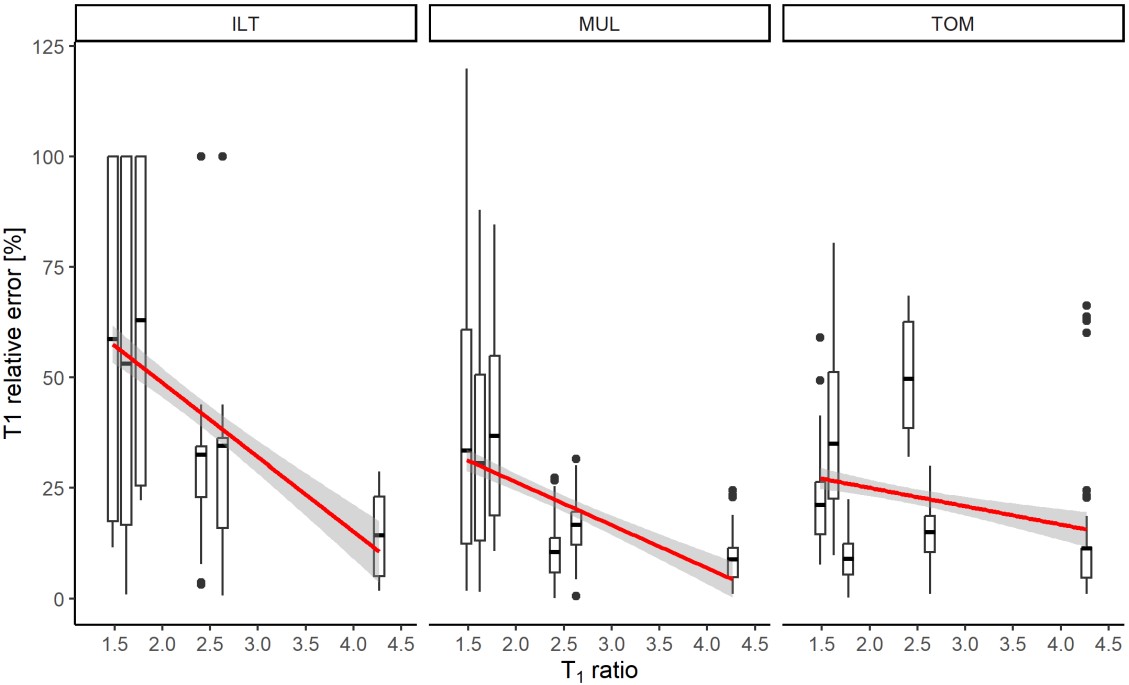

**Fig 1. Dependence of relative error on the $T_1$ ratio for two components.** The red line denotes estimated linear dependence. The relative error was computed for $T_1$ estimated using the inverse Laplace transform (ILT), bounded optimization with multiple starting points (MUL), and TOMer-based bounded optimization (TOM).

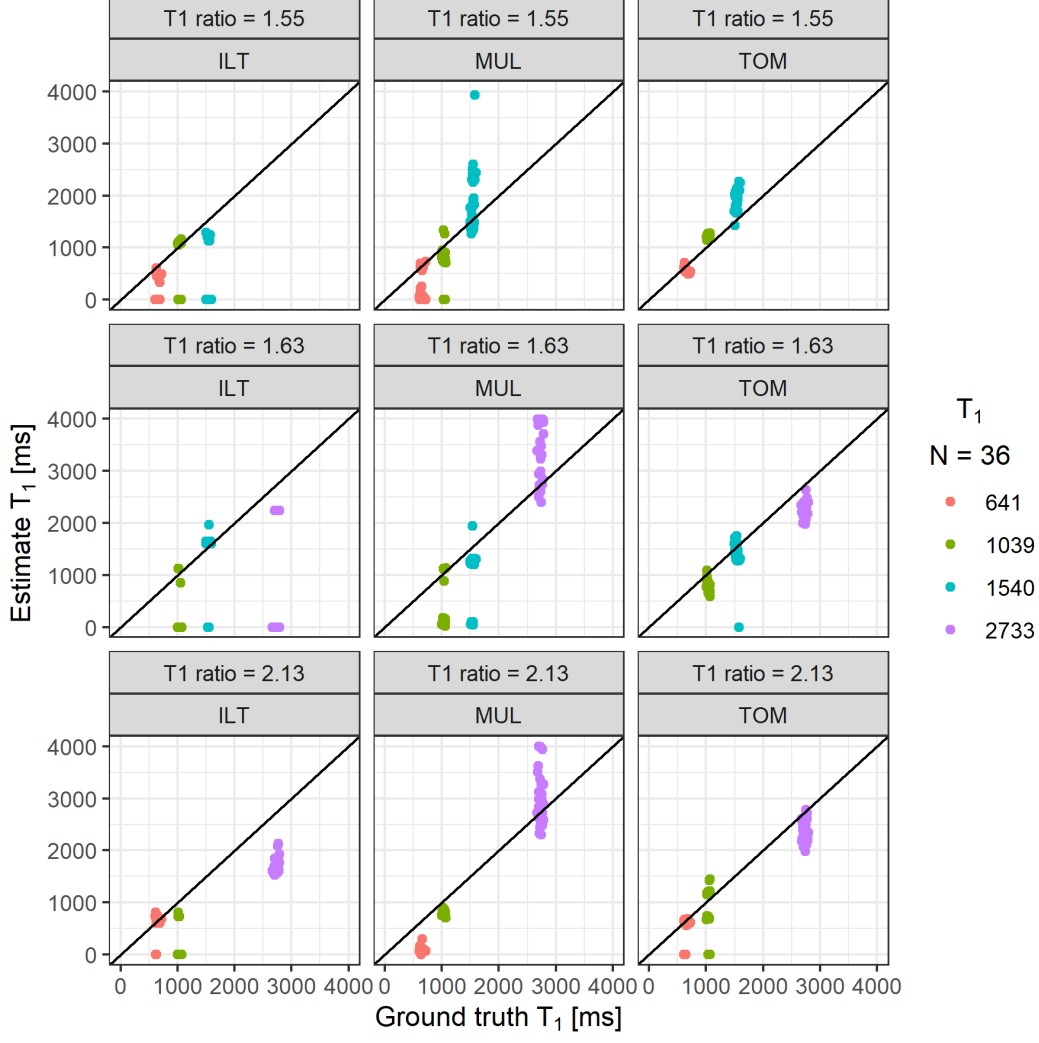

**Fig 2. Estimated $T_1$ constants with their respective ground-truth values stratified according to the mean $T_1$ ratio in data and estimation method.** The relative error was computed for $T_1$ estimated using the inverse Laplace transform (ILT), bounded optimization with multiple starting points (MUL), and TOMer-based bounded optimization (TOM).

The mean relative error of the three-component datasets varied across different $T_1$ ratios and estimation methods (see Fig 2). The ILT method showed improvement in the estimation accuracy with increasing $T_1$ ratio. While no component could be meaningfully assessed (mean relative $A_0$ error higher than 50%) at $T_1$ ratios of 1.55 and 1.63, two components could be evaluated at a $T_1$ ratio of 2.13 (with mean relative error of $T_1$ estimation equal to 20.47% and 38.18%).

The accuracy of MUL and TOM varied across the different $T_1$ ratios. The MUL method achieved the best performance at the $T_1$ ratio of 2.13 (mean relative error of $T_1$ estimation 40.22%), while the TOM method showed the best performance at the $T_1$ ratio of 1.63 (mean relative error of $T_1$ estimation 18.00%). Overall, the mean relative error was highest for the ILT method, lower for the MUL method, and lowest for the TOM method across all examined $T_1$ ratios.

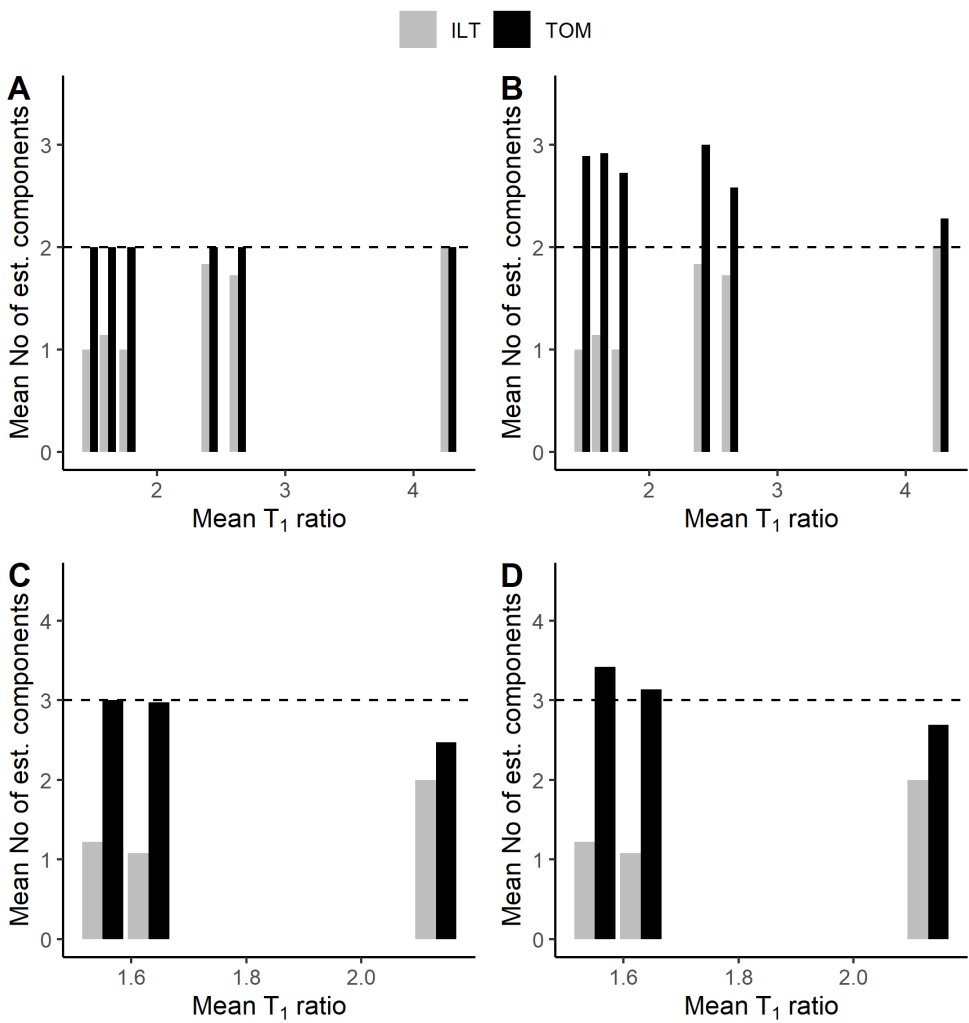

**Fig 3. Mean estimated number of $T_1$ components per voxel, computed across all voxels in the given composite dataset denoted by its mean $T_1$ ratio.** A, B: two-component model; C, D: three-component model. The number of estimated components changes with the introduction of a zero-value component (B, D). ILT – inverse Laplace transform, TOM – TOMer-based bounded optimization.

## Component identification ability

For the two-component models, the ILT method underestimated the expected number of components (1.45 components per voxel on average), while the TOM method estimated their exact number (see Fig 3). When an "artificial" ground-truth component with the value of $A_0$ equal to zero was assumed, the TOM method estimated 2.73 components per voxel on average, while the ILT estimate remained unchanged.

The results were similar for the three-component models. ILT estimated 1.44 components per voxel, and TOM estimated 2.82 components per voxel. The introduction of a zero-component led to a slight overestimation with the TOM method (3.08 components per voxel), while the ILT estimate remained unchanged.

The presence of a zero component affected the $A_0$ estimation accuracy of the TOM method (see Table 4). The average maximum mean error of the $A_0$ estimate increased by 12.19% and decreased by 1.52% for the two-component and three-component models, respectively. In contrast, the ILT method showed improvement, with the average maximum

**Table 4. Average maximum mean error of $A_0$ for two and three-component models with and without the zero-component.**

| Number of all components | Number of non-zero components | ILT – average maximum mean error [%] | TOM – average maximum mean error [%] |
|---|---|---|---|
| 2 | 2 | 51.36 | 44.51 |
| 3 | 2 | 34.25 | 56.70 |
| 3 | 3 | 70.04 | 38.80 |
| 4 | 3 | 52.53 | 37.28 |

ILT – inverse Laplace transform, TOM – TOMer-based bounded optimization.

mean error of the $A_0$ estimate decreased by 17.11% and 17.51%. In absolute terms, the TOM method performed better on the three-component models, while the ILT method performed better on the two-component models.

## Discussion

Successful practical application of multiexponential analysis is a non-trivial problem with many pitfalls [19]. It requires favorable conditions such as a high SNR, knowledge of the decay coefficients' distribution, and their sufficiently high ratio. Even if such conditions are met, the results must be subjected to a certain degree of scrutiny. The spin-lattice relaxation times occurring in cortical gray matter and bordering subcortical white matter generally do not exhibit these properties. Nevertheless, some works have reported successful $T_1$ estimation even in the face of these obstacles [16,20].

We have compared these methods with common methodological approaches used for multiexponential analysis in MRI. The methods were evaluated across the three key degrees of freedom: the number of signal components, the ability to estimate this number, and the ratio of decay constants representing these components. None of the three methods dominated across all degrees of freedom.

The number of components to be estimated is a crucial limiting factor of multiexponential analysis. While two-exponential analysis can generally be performed with reasonable results [21], the performance of most methods decreases when presented with three or more exponential functions [17]. The ILT and MUL methods followed this expected behavior, and their overall mean $T_1$ relative error increased by 10% to 20% when faced with three and four-component signals. The TOM method showed a different trend where accuracy increased by 1% to 2% when faced with additional components. The accuracy of $A_0$ estimation was notably lower than the accuracy of $T_1$ estimation (28.39% average relative error increase) but followed a similar trend. Reasons for this contradictory behavior of the TOM method are hypothesized in the further sections.

In cases where the exact number of components is not known beforehand, a suitable method must be used to estimate it. We have tested two out of the three methods with different approaches of component number estimation. The MUL method was not tested because it currently does not support component number estimation. The TOM method tends to overestimate the true number of components. This overestimation is more severe in the case of the two-component model (2.82 as opposed to 2.00) than in the case of the three-component model (3.08 as opposed to 3.00). This overestimation is likely due to the selection process of the final model. As the SSE naturally decreases with the number of model components, it leads to overestimation of their count. The ILT method tends to underestimate the number of components (1.45 and 1.44 as opposed to 2.00 and 3.00 respectively). This underestimation is directly tied to the ratio of the decay constants and improves as the ratio increases. The estimation accuracy of both methods is also affected by the assumed presence of the zero component. While the accuracy of the TOM method decreased in case of the 2-component signal, it slightly increased in case of the 3-component signal. The accuracy of ILT increased in both cases. This increase is likely a byproduct of the evaluation scheme used.

The performance of each method is expected to increase with the ratio of decay constants [17]. This trend was generally observed for the ILT and MUL methods, but only minimally for the TOM method according to the $R^2$ values of the respective linear fits. We hypothesize that this discrepancy is due to the way in which the TOM method imposes stability for parameter estimation. This method uses optimization settings such as the gradient change and objective function accuracy, which prevent the solution from diverging substantially from the initial starting point. The starting point is deliberately chosen to sample the range of the most commonly occurring relaxation times in gray and white matter at 3 T. Consequently, the optimization tends to yield plausible values of $T_1$ constants, not necessarily because true values were found, but because the optimization stopping criteria were met.

This phenomenon is known as self-bias and has similarly been noted in the context of fitting two exponentials using the Nelder–Mead simplex method [19]. According to the authors, "… if your (first) estimate is close to the solution you are hoping for, this kind of procedure may simply tell you what you want to hear." The performance of the MUL method supports this claim. When one of the multiple starting points evaluated was set according to the scheme used by TOM, worse performance compared to TOM indicated that a different starting point better fit the overall signal.

The results should be considered in the light of some limitations. The final ground-truth dataset was created by combining signals from multiple voxels, effectively performing a cumulation that suppresses noise in the original data. Given the low initial degree of noise and the small number of cumulated signals (two to four), we assume this cumulation has a negligible effect. Amplitudes of the signals from the individual voxels were not modified before their combination, creating combined signals with all components being represented with equal weights. This did not allow for investigation of the effect of their weight's ratio on the estimation accuracy. This effect has been analyzed previously, and the equal distribution of weights yielded the best results [30]. The composite signals were created by an addition in the image domain as opposed to the addition in the frequency domain which occurs in the natural MRI experiment. As the relationship between the frequency and image domain is given by the Fourier transform, which is linear, and thus preserves the operation of addition we do not expect a substantial effect due to this simplification.

The MRI sequence used in our work differs from the sequence reported with the TOM method [20]. As we could not successfully replicate the expected results using the original sequence [16,22], we opted for a different sequence. Both sequences are variations of the conventional IR relaxometry sequence [24] and have been developed for $T_1$ relaxometry. Therefore, we do not expect this change to affect the generalizability of the TOM method substantially.

The discussed models were developed under the assumption of certain favorable conditions, such as complete dependence of the MRI contrast on longitudinal relaxation $T_1$. These conditions might not be fully met in a practical imaging setting, so additional considerations are required when designing and implementing the imaging sequence. The aim of this study was to assess $T_1$ relaxation times corresponding to brain tissue measured at 3 T, but an acquisition protocol with field strength of 9.4 T was used. While the concentrations of the measured liquids ($MnCl_2$) were adjusted accordingly, artifacts related to the increase in the magnetic field's strength could be expected. Visual inspection of the acquired image did not show any substantial artifacts, so their effect was considered negligible.

Improvements to the multiexponential modeling of cortical lamination could be made in a variety of ways. Better estimation accuracy could be achieved by extension of multiexponential analysis to higher dimensions. By tuning the MRI sequence sensitivity to not only the parameter of interest (e.g., $T_1$), but also to additional parameters (such as $T_2$ or diffusion coefficient $D$), stabilization of their estimates could be achieved [31]. The approach has been provided with a statistical theory and additional validations based on Monte-Carlo simulations and phantom measurements [32]. Although the conditioning of the inverse problem has improved, the analysis remains sensitive to experimental noise and decay constant ratio. As the authors used comparatively generous SNRs (40 dB and 33 dB), the performance in more common MRI settings remains to be investigated.

Further improvements in SNR and thus the $T_1$ estimation robustness could be made by signal averaging. At first glance, this would require repeated measurement of the same subject, which would not only prolong the total scan time but also introduce the need for suppression of movement-induced artifacts and registration of images from multiple runs.

Assuming a region-specific homogeneity of cortical lamination, a more feasible form of averaging could involve analyzing $T_1$ decay at the ROI level as opposed to the voxel level. Such profiling has been attempted for brain tumors using $T_2$ relaxation [33]. Using prior knowledge of cortical lamination thickness variance could help in further constraining the underlining optimization problem and thus help in estimating decay constants in proximity. Unfortunately, this knowledge would not be available in the case of pathologies, where the application of medical imaging is the most beneficial.

## Conclusion

Current methodological approaches do not allow for sufficiently precise estimation of $T_1$ decay constants spanning the range of cortical gray matter and bordering subcortical white matter. The lowest relative error achieved across multiexponential models was around 20%. Yet achieving even this level of estimation accuracy requires either $T_1$ ratios that rarely occur in the cerebral cortex or knowledge of the number of relaxation components and their expected values to a degree that is seldom feasible. As the $T_1$ estimation represents only the first step in a longer processing pipeline, additional errors are inevitably introduced in subsequent delineation and visualization steps. Therefore, the $T_1$ estimation would need to achieve substantially lower error—on the order of 5%—to ensure clinically meaningful precision. Visualization of neocortical layers based on these longitudinal relaxation-time estimates is unlikely to represent their true structure. Future research should focus on multiexponential modeling of the allocortical layers due to the more favorable anatomical properties, targeting specific areas, such as entorhinal region, as opposed to whole brain examination.

## Supporting information

**S1 Fig. 3D model of a custom MRI phantom.**
(TIF)

**S2 File. Alignment algorithm description.**
(DOCX)

**S3 File. Modified maximum mean error.**
(DOCX)

**S4 File. Full results of regression analysis.**
(PDF)

**S5 File. Parametric maps of ground truth and estimates.**
(PDF)

**S6 File. Table with mean and standard deviation of ground truth and estimates.**
(PDF)

## Author contributions

**Conceptualization:** Jakub Jamárik, Jiří Vitouš, Radovan Jiřík, Daniel Schwarz.

**Data curation:** Jakub Jamárik.

**Formal analysis:** Jakub Jamárik.

**Investigation:** Jiří Vitouš.

**Methodology:** Jakub Jamárik, Jiří Vitouš, Radovan Jiřík, Daniel Schwarz.

**Software:** Jakub Jamárik.

**Visualization:** Jakub Jamárik.

**Writing – original draft:** Jakub Jamárik.

**Writing – review & editing:** Jiří Vitouš, Radovan Jiřík, Daniel Schwarz, Eva Koriťáková.

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
