## [Decision Letter · Decision Letter 0]

17 Sep 2025

Dear Dr. Koriťáková,

Thank you for submitting your manuscript to PLOS ONE. After careful consideration, we feel that it has merit but does not fully meet PLOS ONE’s publication criteria as it currently stands. Therefore, we invite you to submit a revised version of the manuscript that addresses the points raised during the review process.

We look forward to receiving your revised manuscript.

Kind regards,

Lin Xu

Academic Editor

PLOS ONE

Journal Requirements:

The work on this publication was supported by the Czech Science Foundation (GAČR), URL:  https://gacr.cz/en/, project No. GA23-06957S (JJ and DS) and by the Masaryk University, URL: https://www.muni.cz/en, project No. MUNI/A/1769/2024 (JJ).

3. Thank you for uploading your study's underlying data set. Unfortunately, the repository you have noted in your Data Availability statement does not qualify as an acceptable data repository according to PLOS's standards.

Reviewers' comments:

Reviewer's Responses to Questions

**Comments to the Author**

1. Is the manuscript technically sound, and do the data support the conclusions?

Reviewer #1: Yes

Reviewer #2: Yes

2. Has the statistical analysis been performed appropriately and rigorously?

Reviewer #1: Yes

Reviewer #2: Yes

3. Have the authors made all data underlying the findings in their manuscript fully available?

Reviewer #1: Yes

Reviewer #2: Yes

4. Is the manuscript presented in an intelligible fashion and written in standard English?

Reviewer #1: Yes

Reviewer #2: Yes

Reviewer #1: Dear authors,

Please provide a more concrete conclusion, which should also emphasize the clinical relevance. The lowest relative error achieved across multiexponential models was around 20%, what expectations were at the beginning of the study? If visualization of cortical layers based on these longitudinal relaxation-time estimates is unlikely to represent their true structure, then what clinical importance does this input in future research?

Reviewer #2: The manuscript is well written with enough details for reader to understand and reproduce. The hypothesis is clearly presented, and the data supports the conclusions.

1) A major concern I have is the data were added in the image domain to generate composite signals. In MRI the data (composite signal) is acquired in frequency domain and then transformed to image domain for fitting. I do not expect that this will change the conclusions, but it should be addressed in the discussions.

2) The results are presented in terms of relative error. It would also be useful to show a table listing ground truth T1 and estimated T1 and standard deviation or error based on 2, 3, and 4 component composite data.

3) Equation (2) is valid for 180-degree inversion. Otherwise, there will be additional fitting variable. Was inversion efficiency verified?

4) Line 136 and Table 1: Seven different MNCl2 concentrations were prepared but why table 2 only uses five different T1s to generate composite signal.

5) It would be helpful to show the parameter maps, and ROI.

**Do you want your identity to be public for this peer review?** For information about this choice, including consent withdrawal, please see our Privacy Policy

Reviewer #1: No

Reviewer #2: **Yes: ** Adil Bashir

---

## [Author Response · Author response to Decision Letter 1]

31 Oct 2025

Dear Editorial Office,

the following text constitutes the response to all reviewer and editor comments. The following text is identical with the responses submitted in the document response_to_reviewers_revision-01.docx.

Response to Editors’ comments

Point 1: PLOS ONE's style requirements

Response 1: We have revised the manuscript, figures, and tables and updated all documents according to PLOS ONE's style requirements.

Point 2: Financial disclosure

Response 2: We have included the amended Role of Funder statement, stating: "The funders had no role in study design, data collection and analysis, decision to publish, or preparation of the manuscript.", in our cover letter. We have also disclosed additional funding information for two authors, who did not provide this information during the initial submission. As this is not possible to amend in the submission system, we have included the complete up to date funding information in the cover letter.

Point 3: Acceptable data repository

Response 3: We have changed the data repository to Zenodo as per PLOS ONE recommended repositories.

Point 4: Data availability statement

Response 4: We have uploaded the data to Zenodo and locked the data access pending acceptance of the manuscript for publication. The data can be accessed from https://doi.org/10.5281/zenodo.17456443.

Point 5: Recommendations to cite specific literature

Response 5: No recommendations to cite specific literature were made by the reviewers.

Point 6: Reference list

Response 6: We have reviewed the reference list ensuring that it is complete and up to date. No references were retracted.

Response to Reviewer 1 comments

Dear reviewer, thank you for your valuable comments and insights, which helped us improve our manuscript.

Point 1: Dear authors, please provide a more concrete conclusion, which should also emphasize the clinical relevance. The lowest relative error achieved across multiexponential models was around 20%, what expectations were at the beginning of the study? If visualization of cortical layers based on these longitudinal relaxation-time estimates is unlikely to represent their true structure, then what clinical importance does this input in future research?

Response 1: At the outset of our experiments, we anticipated that the achievable precision of multiexponential decomposition would be limited by the ill-posed nature of the inverse problem and by the relatively narrow range of T₁ values encountered in the brain cortex. The lowest relative error of approximately 20% observed in our results was consistent with our expectations and with the theoretical limits of the method. However, as the modeling represents only the first step in a longer processing pipeline, additional errors are inevitably introduced in subsequent delineation and visualization steps. Therefore, the modeling itself would need to achieve substantially lower error—on the order of 5%—to ensure clinically meaningful precision, underscoring why the multiexponential approach at 3 T is unlikely to yield anatomically valid layer maps.

Clinical importance of our results is twofold. First, as our findings give estimates on the expected precision which can be achieved via multiexponential modeling, they will help clinical researchers better assess the credibility of any published research based on this methodological framework and evaluate whether to employ it in their own research. Second, as the estimation accuracy decreases with the number of expected cortical layers our findings suggest that the multiexponential approach is more suitable for examination of allocortex as opposed to the neocortex. Whether this could form a basis for successful clinical applications, for example for the early detection of Alzheimer’s disease via in vivo examination of entorhinal regions, remains to be seen in future research.

We have updated the conclusion to better reflect the above-mentioned and as such to deliver a more concrete message (line 429-436).

Response to Reviewer 2 comments

The manuscript is well written with enough details for reader to understand and reproduce. The hypothesis is clearly presented, and the data supports the conclusions.

Dear reviewer, thank you for reading our manuscript and providing us with your positive and encouraging feedback. We believe that after the changes made to address yours and Reviewer 1`s comments, the manuscript is now even better.

Point 1: A major concern I have is the data were added in the image domain to generate composite signals. In MRI the data (composite signal) is acquired in frequency domain and then transformed to image domain for fitting. I do not expect that this will change the conclusions, but it should be addressed in the discussions.

Response 1: We have addressed the issue of addition in the composite domain as opposed to frequency domain in the discussion (line 388-391). As we have employed Cartesian sampling the relationship between frequency and image domains is given by the Fourier transform which is a linear operator thus preserving the operation of addition. We therefore believe that our simplification does not significantly hamper generalizability of our experiment.

Point 2: The results are presented in terms of relative error. It would also be useful to show a table listing ground truth T1 and estimated T1 and standard deviation or error based on 2, 3, and 4 component composite data.

Response 2: We have added a table with mean and standard deviation for ground truth values and parameter estimates, for all composite data sets, to the supplementary materials, as this table is rather large (see supplementary material S6, referenced in manuscript text on line 272-274).

Point 3: Equation (2) is valid for 180-degree inversion. Otherwise, there will be additional fitting variable. Was inversion efficiency verified?

Response 3: The inversion pulse was automatically set and verified by the scanner, with the phantom placed in the isocenter to further ensure maximal inversion efficiency. We have added the preceding statement to the methods section of the manuscript (line 148-149) to make this information clear to the reader.

Point 4: Line 136 and Table 1: Seven different MNCl2 concentrations were prepared but why table 2 only uses five different T1s to generate composite signal.

Response 4: This is due to the low signal quality and therefore unreliable ground-truth T1 estimate of the remaining ROI (this fact is mentioned in the methods sections titled Ground truth – line 160-161) which prevented us from making additional meaningful ROI combinations. We have added clarification to Table 1. caption to make this fact more easily available to the reader (line 174-175).

Point 5: It would be helpful to show the parameter maps, and ROI.

Response 5: We have added ground truth and parametric estimate maps for the whole image and the individual ROI to the supplementary materials (see supplementary material S5, referenced in manuscript text on line 272-274). As each ground truth constitutes a 6 x 6 square extracted from a mostly homogeneous area and the parameter estimates can vary substantially, each image must be interpreted w. r. t. to its individual pseudocolor scale.

---

## [Decision Letter · Decision Letter 1]

17 Nov 2025

Limitations of multiexponential T1 mapping of cortical myeloarchitecture

PONE-D-25-37777R1

Dear Dr. Koriťáková,

We’re pleased to inform you that your manuscript has been judged scientifically suitable for publication and will be formally accepted for publication once it meets all outstanding technical requirements.

Kind regards,

Lin Xu

Academic Editor

PLOS ONE

Additional Editor Comments (optional):

Reviewers' comments:

Reviewer's Responses to Questions

**Comments to the Author**

Reviewer #1: All comments have been addressed

Reviewer #2: All comments have been addressed

2. Is the manuscript technically sound, and do the data support the conclusions?

Reviewer #1: Yes

Reviewer #2: Yes

3. Has the statistical analysis been performed appropriately and rigorously?

Reviewer #1: Yes

Reviewer #2: Yes

4. Have the authors made all data underlying the findings in their manuscript fully available?

Reviewer #1: Yes

Reviewer #2: Yes

5. Is the manuscript presented in an intelligible fashion and written in standard English?

Reviewer #1: Yes

Reviewer #2: Yes

Reviewer #1: Dear authors,

Congratulations on your work.

The paper "Limitations of multiexponential T1 mapping of cortical myeloarchitecture" has been improved.

Reviewer #2: The hypothesis is clearly stated, and results supports the conclusions. The authors have satisfactorily addressed all concerns in the revised version.

**Do you want your identity to be public for this peer review?** For information about this choice, including consent withdrawal, please see our Privacy Policy

Reviewer #1: No

Reviewer #2: **Yes: ** Adil Bashir

---

## [Editor Report · Acceptance letter]

PONE-D-25-37777R1

PLOS ONE

Dear Dr. Koriťáková,

I'm pleased to inform you that your manuscript has been deemed suitable for publication in PLOS ONE. Congratulations! Your manuscript is now being handed over to our production team.

Kind regards,

on behalf of

Dr. Lin Xu

Academic Editor

PLOS ONE